# In Vivo Gastroprotective and Antidepressant Effects of Iridoids, Verbascoside and Tenuifloroside from *Castilleja tenuiflora* Benth

**DOI:** 10.3390/molecules24071292

**Published:** 2019-04-02

**Authors:** Ricardo López-Rodríguez, Maribel Herrera-Ruiz, Gabriela Trejo-Tapia, Blanca Eda Domínguez-Mendoza, Manasés González-Cortazar, Alejandro Zamilpa

**Affiliations:** 1Centro de Desarrollo de Productos Bióticos, Instituto Politécnico Nacional, Col. San Isidro, Yautepec, Morelos C.P. 62731, Mexico; richard_lorr@hotmail.com; 2Centro de Investigación Biomédica del Sur, Instituto Mexicano del Seguro Social, Argentina No. 1, Col. Centro, Xochitepec, Morelos C.P. 62790, Mexico; cibis_herj@yahoo.com.mx (M.H.-R.); gmanases2000@gmail.com (M.G.-C.); 3Centro de Investigaciones Químicas, Universidad Autónoma del Estado de Morelos, Av. Universidad 1001, Col. Chamilpa, Cuernavaca, Morelos C.P. 62209, Mexico; bed@uaem.mx

**Keywords:** *Castilleja tenuiflora* benth., antidepressant, gastroprotective, cold-restraint stress, tenuifloroside, verbascoside

## Abstract

Stress is an important factor in the etiology of some illnesses such as gastric ulcers and depression. *Castilleja tenuiflora* Benth. (Orobanchaceae) is used in Mexican traditional medicine for the treatment of gastrointestinal diseases and nervous disorders. Previous studies indicated that organic extracts from *C. tenuiflora* had gastroprotective effects and antidepressant activity. In this study, we aimed to evaluate the gastroprotective and antidepressant activity of fractions and isolated compounds from the methanolic extract (MECt) of *C. tenuiflora* in stressed mice. Chromatographic fractionation of MECt produced four fractions (FCt-1, FCt-2, CFt-3, and FCt-4) as well as four bioactive compounds which were identified using TLC, HPLC and NMR analyses. The cold restraint stress (CRS)-induced gastric ulcer model followed by the tail suspension test and the forced swim test were used to evaluate the gastroprotective effect and antidepressant activity of the extract fractions. FCt-2 and FCt-3 at 100 mg/kg had significant gastroprotective and antidepressant effects. All isolated compounds (verbascoside, teniufloroside and mixture geniposide/ musseanoside) displayed gastroprotective effects and antidepressant activity at 1 or 2 mg/kg. The above results allow us to conclude that these polyphenols and iridoids from *C. tenuiflora* are responsible for the gastroprotective and antidepressant effects.

## 1. Introduction

Stress is a main factor in the etiology of some ailments such as gastric ulcer (GU) and depression suggesting a relationship between these two ailments [1,2]. In fact, this comorbidity has been established by a link between cholinergic hypersensitivity and hypothalamic-pituitary-adrenal axis (HPA) activation by chronic stress [3]. The HPA axis is activated concurrently with alterations in neurotransmitter metabolism (e.g., serotonin, dopamine, norepinephrine), which are thought to be primarily responsible for the psychological and behavioral symptoms [4]. Gastroprotective and antidepressant drugs may be used indistinctly for the treatment of both ailments: gastric ulcer and depression [2]. Pre-clinical studies have shown that antidepressant drugs, such as imipramine [5], duloxetine, amitriptyline, fluoxetine and mirtazapine are able to benefit mice with ulcers showing an antidepressant effect [6].

Medicinal herbs are considered an excellent source of useful compounds in the treatment of several diseases, since they frequently contain secondary metabolites that have multitarget activities [7,8]. Mexican traditional utilizes a wide variety of plant species that might be important sources of compounds with gastroprotective and antidepressant activities [9]. *Castilleja tenuiflora* Benth. (Orobanchaceae) is a small perennial herb which is found in disturbed areas, temperate pine-oak forests in Mexico [10,11]. This species, known in Nahuatl as “Atzoyatl” is used medicinally to relieve symptoms related to stomach and central nervous system symptoms including gastrointestinal and nervous disorders [12]. In addition, the infusion of this medicinal plant is useful for the relief of cough, vomiting, dysentery and liver pain [13]. Previous reports indicate that organic extracts from aerial parts of *C. tenuiflora* had anti-ulcerogenic and antidepressant effects [14,15].

Furthermore, this specie contains a wide variety of compounds (Figure 1), such as: glycosylated iridoids (e.g., aucubin, bartsioside, geniposide (**GP**), geniposidic acid, mussaenoside (**MS**), methyl 8-epiloganin, shanzhiside and caryoptoside), phenolic compounds (e.g., phenylethanoid derivatives verbascoside (**VB**), isoverbascoside, tenuifloroside (**TN**)), and the flavonoids (e.g., apigenin, luteolin-5-methyl ether and quercetin glycosides) [14,15,16,17,18]. Considering the previously demonstrated biological activity of several extracts of *C. tenuiflora*, the aim of this work was to evaluate the gastroprotective and antidepressant activity of fractions and isolated compounds from the methanolic extract of *C. tenuiflora* in stressed mice.

## 2. Results

### 2.1. Chemical Identification of Isolated Compounds

Previous studies of *Castilleja tenuiflora* Benth., described the isolation and identification of tenuifloroside, verbascoside and a mixture of iridoids (geniposide and mussaenoside) as the major constituents. In this study the presence and quantities of these compounds in the methanol extract (MECt) and fractions (FCt-2, FCt-3, FCt-4) were analyzed by HPLC (Figure 2) and by comparison with the previously described spectroscopic NMR data [14,15]. The quantities of these polyphenols and iridoids contained in each treatment are described in Table 1. In all cases, verbascoside was the major compound.

Table 1 also, shows the quantity of these compounds in MECt and FCt-2, FCt-3, FCt-4, where verbascoside was the major compound in all treatments

### 2.2. Gastroprotective and Antidepressant Activity

#### 2.2.1. Effects of Castilleja Pretreatment on CRS-Induced Gastric Damage

The gastroprotective effect of all treatments is shown in Figure 3. While the negative control group (VEH, non-treatment), had hyperemia and mucosal erosions (Figure 3B), these kinds of macroscopic lesions were not observed in the normal control group (Basal, Figure 3A), and the ulcerated area in the VEH group was significantly (*p* ≤ 0.05) higher than the basal level (Table 2). This confirms that repeated exposure to cold (hours-days) causes stress damage in mice, and that CRS produces gastric mucosal ulceration in rats. In general, these lesions are caused by increased acid secretion, inhibition of gastric mucosal prostaglandin synthesis and bicarbonate secretion, disruption of the gastric mucosal barrier and decrease of gastric mucosal blood flow [19]. Since mucus has a critical role in protecting the stomach and enhancing healing of the stomach walls, the CRS model is used to evaluate mucosal and cytoprotective agents [20]. The antidepressant and gastroprotective drugs imipramine (15 mg/kg) and misoprostol (0.05 mg/kg) reduced the formation of mucosal lesions (Figure 3C,D) and significantly decreased the ulcerated area. Previous studies indicated that misoprostol displayed cytoprotective effect producing a stimulation of bicarbonate and mucus secretion and increment of mucosal blood flow [21]. On the other hand imipramine displayed gastroprotective activity via multiple mechanisms including antisecretory, mucogenic and antioxidant activities [6].

Likewise, the MECt, FCt-2 and FCt-3 treatments decreased the incidence of mucosal lesions with inhibition percentages of 97.6, 92.9 and 99.4% respectively; the FCt-4 treatment was less active (61.7%) and greater lesions and hyperemia were observed in the mucosa (Table 2 and Figure 3E–H). All isolated compounds displayed gastroprotective effects at doses of 1 and 2 mg/kg. The GP/MS mixture showed an inhibition of 99.8 and 99.7, VB of 99.8 and 99.6 and TN of 99.3 and 98.9%, at the 1 and 2 mg/kg doses, respectively (Figure 3I–N and Table 2). These results correlate with percentage of lesion area and ulcer index. VEH group displayed values of 23.01 ± 8.3% and 16.1 ± 1.6, respectively (Table 2).

These data are similar to the anti-ulcerogenic effect produced by an ethyl acetate extract of this species in an ethanol-induced gastric ulcer model [14]. This is the first time that gastroprotective effect of iridoids (GP/MS) and tenuifloroside is reported. In the case of iridoids, this effect could be related to the antioxidant capacity reported for GP [22]. Also, tenuifloroside is a lignan like syringaresinol, which has been described to have an inhibitory effect on gastric ulcers [23]. Verbascoside (40 mg/kg) has been reported to have anti-ulcerogenic activity through the neutralization of acid and gastric secretion by inhibiting H^+^/K^+^-ATPase [24]. This effect was related to the antioxidant, neuroprotective, antiinflammatory and cytoprotective activities of this phenylpropanoid [25]. It has been documented that chronic stress is the main risk factor for many diseases such as gastric ulcer and neuropsychiatric disorders [26], playing an important role in depression by activating the hypothalamic-pituitary adrenal axis and the sympathetic nervous system [1].

#### 2.2.2. Antidepressant Effect of Integrate Extract and Fractions

The antidepressant activity of control drugs versus the basal level and VEH is shown in Figure 4. The results of the tail suspension test (TST; Figure 4A) and forced swim test (FST; Figure 4B) indicate that both positive controls IMI (15 mg/kg) and MISO (0.05 mg/kg) significantly decreased immobility time to a level similar to the basal group, while the VEH group (*p* < 0.05); these comparisons validate the pharmacological model. These data also suggest that CRS induced increase in immobility time in FST and TST, though TST is more sensitive for detection of antidepressant-like activity.

Similar to imipramine (a tricyclic antidepressant monoamine), MISO was effective in both behavioral models (FST and TST); this is the first time misoprostol, a gastroprotective drug, 0.05 mg/kg), is described as antidepressant. However, this medicine did not have this antidepressant effect at higher dose (0.32 mg/kg) [27]. It is possible that at the higher dose, MISO produces a sedative effect like prostaglandins E which decrease the locomotor and exploratory activities of several animal species [28]. In fact, a recent study demonstrated that misoprostol (0.050 mg/kg), exerts an anxiolytic effect in mice with aspirin-induced gastric ulcers [29].

The antidepressant activity of the MECt and its fractions are described in Figure 5. MECt and fractions FCt-2 and FCt-3 decreased the duration of immobility time in TST similar to IMI and MISO. However, FCt-4 did not present a significant difference in immobility time in TST when compared to VEH (Figure 5A). In the FST, fractions FCt-2, FCt-3 and FCt-4 significantly decreased the duration of immobility time compared with VEH. In this case, the integrated extract MECt did not decrease this parameter (Figure 5B).

Although TST and FST share theoretical bases, there are differences between them; TST prevents problems of hypothermia or motor dysfunction that could interfere with performance in the swimming test, while FST could overcome the tail-climbing problem in TST [30]. The chemical analysis showed that the only difference between fractions FCt-2 and FCt-3 is the concentration of GP/MS, VB and TN (Table 1). *C. tenuiflora* is traditionally used to treat nervous conditions. Recently, antidepressant, sedative and hypnotic effects of MECt were reported, in which this extract induced a decrease of the motor parameters [15]. However, in this sub-chronic model the antidepressant effect was observed only in the TST. This variation could be due to the chronicity of this stress model. The mechanisms that underlie the “chronicity switch” from acute to chronic time domains lies at the crux of understanding stress-related pathologies [31].

#### 2.2.3. Antidepressant Effect of Isolated Compounds

The antidepressant effect of GP/MS, VB and TN are shown in Figure 6. Verbascoside (VB at 2 mg/kg), was the unique isolated compound that significantly decreased immobility time in the TST (Figure 6A) and FST (Figure 6B), similar to the two drug group (IMI and MISO). Liang et al. [32] suggest that this phenyl ethanoid could be a promising drug for treating illnesses related to the central nervous system. Previous studies have shown that VB (3 mg/kg) exerts a sedative effect [33]. In the case of the mixture of iridoids (GP/MS) at 1 and 2 mg/kg, these significantly decrease immobility time compared to the VEH group (Figure 6B) in the FST (Figure 6B), but not in the TST (Figure 6A). Liu et al. [34] have shown that GP (10 mg/kg) exerts an antidepressant effect (evaluated by the FST and TST); due to increasing the levels of 5-hydroxytryptamine. Finally, TN displayed antidepressant effects only in the FST model at the lowest dose (1 mg/kg, Figure 6B). Similar to VB, this is the first time that tenuifloroside is reported as an antidepressant. Sesamin, a lignan that is structurally similar to TN, has been shown to improve depression induced by chronic stress in mice; in that case, the biological effect was related to a decrease in levels of 5-HT and NE [35]. Other lignans, such as bisepoxylignan glycoside and simplexoside, are able to act as depressants and antidepressants of the CNS [36,37]. These effects could be due to chemical differences between lignan types [38].

## 3. Materials and Methods

### 3.1. General Information

All NMR spectra and two-dimensional spectroscopy experiments COSY, HSQC, HMBC were recorded on an INOVA-400 instrument (Varian, Palo Alto, CA, USA) at 400 MHz for ^1^H-NMR spectra in CDCl_3_, CD_3_OD or DMSO-d6 with tetramethylsilane (TMS) as internal standard. Chemical shifts are reported in δ values. Analytical TLC was carried out on precoated Merck silica gel 60F254 or RP-18F254 plates. 

### 3.2. Extraction and Fractionation of C. Tenuiflora

Aerial parts of *C. tenuiflora* were collected in Juchitepec, Mexico State, Mexico in December 2015 (2800 m.a.s.l., latitude 19°10 N longitude 98°92 W) and identified by Rolando Ramirez, M. Sc., Head Curator of the HUMO Herbarium of UAEM, Morelos, Mexico with reference voucher HUMO25205.

Plant material was dried at room temperature in the dark, milled (4–6 mm) and extracted by maceration with methanol (2 kg, 6 L, 24 h). The liquid extract was concentrated under reduced pressure conditions in a rotary evaporator (Büchi-490; BUCHI Labortechnik AG, Flawil, Switzerland) at 40 °C (215 g, yield: 10.7%). The methanol dried extract (MECt, 50 g) was fractionated by silica gel open column chromatography (300 g, 20 × 60 cm). The mobile phase consisted of a dichloromethane/acetone/ methanol gradient system; CH_2_ Cl_2_ (100), CH_2_Cl_2_:CH_3_COCH_3_ (50/50), CH_3_COCH_3_ (100), CH_3_OH (100). 100 mL samples were collected to give 4 fractions (FCt-1, 0.85 g; FCt-2, 13.7 g; FCt-3, 27.7 g and FCt-4, 1.7 g). Since the less polar fraction (FCt-1) was mainly composed of fatty acids, it was not considered for pharmacological analysis. While the GP/MS mixture was obtained from FCt-2, chemical separation of FCt-3 and FCt-4 allowed for the isolation of VB and TN. This purification process was done following the chromatographic method previously described. All the compounds were identified based on their spectroscopic data (For spectra see Appendix A), which were compared with those previously found in the literature [15,16].

Geniposide (**GP**): White amorphous powder, the spectrum UV displayed absorption at λ_max_: 240 nm; ^1^H-NMR (400 MHz, methanol-d: CDCl_3_): δ 4.95 (1H, d, 7.8 Hz, H-1), 7.45 (1H, s, br, H-3), 3.11–3.16 (1H, m, H-5), 2.03 (1H, ddd, 1.9, 8.6, 16 Hz, H-6a), 2.82 (1H, dd, br, 8.6, 16.4 Hz H-6b), 5.79 (1H, s, H-7), 2.62 (1H, dd, 7.8, 7.8, Hz, H-9), 4.25 (1H, d br, 14 Hz, H-10a, H-10b), 4.68 (1H, d, 7.8 Hz, H-1′), 3.2-3.3 (1H, m, H-2′), 3.36 (1H, m, H-3′), 3.13 (1H, dd, 8, 9.2 Hz, H-4′), 3.64 (1H, m, H-5′), 3.7 (1H, m, H-6a′), 3.78 (1H, dd, 2.3, 12 Hz, H-6b′), 3.68 (3H, s, OCH_3_); ^13^C-NMR (100 MHz, Methanol-d: CDCl_3_): δ 98.3(C-1), 150.9 (C-3), 112.7 (C-4), 35.9 (C-5), 39.1 (C-6), 128.6 (C-7), 143.4 (C-8), 46.1 (C-9), 60.8 (C-10), 168.5 (C-11), 99.7 (C-1′), 73.5 (C-2′), 76.8 (C-3′), 70.5 (C-4′), 76.8 (C-5′), 62.1 (C-6′), 51.5 (OCH_3_).

Mussaenoside (**MS**): White amorphous powder, the spectrum UV displayed absorption at λ_max_: 238 nm; ^1^H-NMR (400 MHz, methanol-d:CDCl_3_): δ 5.38 (1H, d, 3.5 Hz, H-1), 7.33 (1H, s br, H-3), 3.24–3.26 (1H, m, H-5), 2.25 (1H, ddd, 10.1, 8.2, 3.5 Hz, H-6a), 1.61–1.69 (1H, m, H-6b), 1.62–1.67 (1H, m, H-7a), 1.69–1.72 (1H, m, H-7b), 2.25 (1H, dd, 10.5, 3.5 Hz, H-9), 1.24, (3H, s, H-10), 4.62 (1H, d, 7.8 Hz, H-1′), 3.2–3.3 (1H, m, H-2′), 3.36 (1H, m, H-3′), 3.13 (1H, dd, 8, 9.2 Hz, H-4′), 3.64 (1H, m, H-5′), 3.7 (1H, m, H-6a′), 3.78 (1H, dd, 2.3, 12 Hz, H-6b′), 3.66 (3H, s, OCH_3_). ^13^C-NMR (100 MHz, Methanol-d: CDCl_3_): δ 94.4 (C-1), 152.4 (C-3), 111.6 (C-4), 30.2 (C-5), 29.9 (C-6), 40.5 (C-7), 79.4 (C-8), 51.6 (C-9), 23.9 (C-10), 168.6 (C-11), 98.9 (C-1′), 73.6 (C-2′), 76.7 (C-3′), 69.9 (C-4′), 76.6 (C-5′), 61.5 (C-6′), 51.3 (OCH_3_).

Tenuifloroside (**TN**): White amorphous powder, the spectrum UV displayed absortion at λ_max_: 211, 228.6 and 282.8 nm; ^1^H-NMR (400 MHz, DMSO-*d_6_*): 6.91 (1H, d, 1.6 Hz, H-2), 7.07 (1H, d, 8.4 Hz, H-5), 6.82 (1H, dd, 8.4, 1.6 Hz, H-6), 4.6 (1H, d, 4.8 Hz, H-7), 2.97 (1H, m, H-8), 4.13 (1H, m, H-9a), 3.74 (1H, dd, 4.0, 4.0 Hz, H-9b), 6.89 (1H, d, 1.2 Hz, H-2′), 6.83 (1H, d, 8.4 Hz, H-5′), 6.83 (1H, dd, 8.4, 1.2 Hz, H-6′), 4.6 (1H, 4.8 Hz, H-7′), 2.97 (1H, m, H-8′), 4.13 (1H, m, H-9a′), 3.74 (1H, dd, 4.0. 4.0 Hz, H-9b′), 4.82 (1H, d, 7.6 Hz, H-1″), 3.24 (1H, m, H-2″), 3.24 (1H, m, H-3″), 3.14 (1H, m, H-4″), 3.49 (1H, m, H-5″), 3.90 (1H, d br, 10.4 Hz, H-6α″), 3.53 (1H, dd, 11.6, 6.4 Hz, H-6β″), 4.13 (1H, m, H-1′″), 2.93 (1H, m, H-2′″), 3.04 (1H, m, H-3′″), 3.24 (1H, m, H-4′″), 2.90 (1H, m, H-5α′″), 3.64 (1H, dd, 5.6, 11.2 Hz, H-5β′″) 3.74 (3H, s, OCH_3_), 5.95 (2H, s, OCH_2_O). ^13^C-NMR (100 MHz, methanol-d): δ 135.59 (C-1), 110.87 (C-2), 149.24 (C-3), 146.2 (C-4), 115.88 (C-5), 118.82 (C-6), 85.4 (C-7), 54.13 (C-8), 71.45 (C-9), 135.91 (C-1′), 107 (C-2′), 146.91 (C-3′), 147.84 (C-4′), 108.47 (C-5′), 118.85 (C-6′), 85.3 (C-7′), 54.17 (C-8′), 71.45 (C-9′), 110.62 (C-1″), 73.59 (C-2″), 77.06 (C-3″), 69.98 (C-4″), 73.35 (C-5″), 68.47 (C-6″), 104.12 (C-1′″), 73.81 (C-2′″), 76.89 (C-3′″), 69.98 (C-4′″), 65.98 (C-5′″), 56.14 (OCH_3_), 101.32 (OCH_2_O).

Verbascoside (**VB**): Brown amorphous powder, the spectrum UV displayed absortion at λ_max_: 220, 248 and 331 nm; ^1^H-NMR (400 MHz, methanol-d): δ 6.753 (1H, d, 1.56 Hz, H-2), 6.73 (1H, d, 8.2 Hz, H-5), 6.62 (1H, dd, 7.81, 1.56 Hz, H-6), 2.84 (1H, m, H-7a), 2.86 (1H, m, H-7b), 3.78 (1H, ddd, 9.5, 7.6, 3.9 Hz, H-8a), 4.1 (1H, ddd, 8.2, 7.0, 3.1 Hz, H-8b), 7.11(1H, d, 1.9 Hz, H-2′), 6.84 (1H, d, 8.2 Hz, H-5′), 7.01 (1H, dd, 7.8, 1.9 Hz, H-6′), 7.65 (1H, d, 16.0 Hz, H-7′), 6.33 (1H, d, 16.0 Hz, H-8′), 4.43 (1H, d, 7.8 Hz, H-1″), 3.45 (1H, dd, 8.9, 8.2 Hz, H-2″), 3.87 (1H, t, 9.3 Hz, H-3″), 4.99 (1H, m, H-4″), 3.56 (1H, m, H-5″), 3.58 (1H, m, H6a″), 3.68 (1H, d, 10.1 Hz H6b″), 5.25 (1H, d, 1.1 Hz, H-1′″), 3.98 (1H, dd, 4.29, 2.1 Hz, H-2′″), 3.63 (1H, dd, 9.56, 3.31 Hz, H-3’”), 3.33 (1H, t, 9.76, 8.59 Hz, H-4′″), 3.61 (1H, m, H-5′″), 1.15 (3H, d, 6.2 Hz, H6′″). ^13^C-NMR (100 MHz, methanol-d): δ 130.1(C-1), 115.12 (C-2), 144.71 (C-3), 143.24 (C-4), 115.7 (C-5), 119.87 (C-6), 35.15 (C-7), 70.83 (C-8), 126.26 (C-1′), 113.85 (C-2′), 145.41 (C-3′), 148.37 (C-4′), 114.92 (C-5′), 121.8 (C-6′), 146.61 (C-7′), 113.31 (C-8′), 166.9 (C-9′), 102.79 (C-1″), 74.79 (C-2″), 80.25 (C-3″), 69.19 (C-4″), 74.61 (C-5″), 60.97 (C-6″), 101.6 (C-1′″), 70.94 (C-2′″), 70.66 (C-3′″), 72.4 (C-4′″), 69.0 (C-5′″), 17.04 (C-6′″).

### 3.3. Drug and C. Tenuiflora Treatments

Imipramine hydrochloride (IMI, >99%, Sigma-Aldrich, St. Louis, MO, USA) and misoprostol (MISO, 95%, Serral, Mexico, CDMX) were used as control antidepressant and gastroprotectant drugs, respectively. All *C. tenuiflora* treatments, Tween 20 (TW 1%, vehicle, Merck, Kenilworth, NJ, USA) and MISO (0.05 mg/kg) were administrated by oral pathway (o.p) and only imipramine (15 mg/kg) was intraperitonially (i.p.). MECt at 500 mg/kg, fractions (FCt-2, FCt-3 and FCt-4 at 100 mg/kg) and isolated compounds (GP/MS, VB and TN) were administered at doses of 1 and 2 mg/kg, and the negative control received only 1% Tween solution. A group of non-stressed animals (BASAL) was used as the baseline control.

### 3.4. Animals

The experiments were standardized and performed on male albino ICR mice (body weight range 38–45 g, Harlan, Mexico City, Mexico). All mice were kept in polypropylene boxes, specific for the vivarium, at a temperature of 24 ± 1 °C, with food (Labina, Purina^®^, St. Louis, MO, USA) and water ad libitum and maintained in a light/dark cycle of 12 h. All tests were performed on 6 mice per group, which were acclimatized to the experiment site for 24 h prior to the experiment. The experimental protocol was previously registered at the Instituto Mexicano del Seguro Social, Mexico (IMSS, R-2018-2103-001) in compliance with the Official Mexican Regulation (NOM-062-ZOO-1999).

### 3.5. HPLC Analysis

The qualitative and quantitative analyses of GP/MS, VB and TN were performed using an Alliance 2695 separation module system (Waters, Milford, MA, USA) coupled with a Spectral System UV2996 PDA detector. Chromatographic separation was carried out using a Supelcosil^TM^ LC-F column (4.6 mm × 250 mm, 5 µm, Sigma-Aldrich, Bellefonte, PA, USA). The mobile phase consisted of two solvent reservoirs: A (Trifluoroacetic acid–water, 0.5%, *v*/*v*) and B (acetonitrile). The gradient system was as follows: 0–1 min, 100–0% B; 2–3 min, 95–5% B; 4–20 min, 70–30% B; 21–23 min, 50–50%, 24–25 min, 20–80% B, 26–27 min, 0–100% B and 28–30 min 100–0% B. The flow rate was 0.9 mL/min with an injection volume of 10 µL. The photo diode array detector wavelength was set at 240, 330 and 280 nm for the identification and quantification of GP/MS, VB and TN, respectively. Calibration curves for GP/MS, VB and TN were prepared by injecting ascendant concentrations (25, 50, 100, 200 and 400 µg/mL) of previously isolated compounds. Analytical parameters of linearity, limit of detection and limit of quantification were measured for verbascoside (y = 6831.7x – 1309.7 R^2^ = 0.9985, LOD = 0.16 μg/mL, LOQ = 0.49 μg/mL); tenuifloroside (y = 9512.9x – 11296, R^2^ = 0.9959, LOD = 0.78 μg/mL, LOQ = 2.36 μg/mL) and GP/MS (y = 5794.6x – 116127 R^2^ = 0.9991, LOD = 0.94 μg/mL, LOQ = 2.84 μg/mL).

### 3.6. Pharmacological Experiments

#### 3.6.1. Cold Restraint Stress-Induced Gastric Ulcer Test (CRS)

This pharmacological model was carried out using the method described by Das and Banerjee [39] with the following adaptations: mice were placed in immobilizing boxes which were stored at 4 ± 2 °C with air circulation, for 30 min (first day). The experimental design described in Figure 7 indicates the exposition to cold restraint stress and administration of treatments, as follow: Time of exposure to CRS was increased by 10 min daily until day four; after this period all mice were deprived of food for 12 h prior to exposure. On the fifth day, the CRS was maintained for 90 min. The administration of each treatment was initiated in the third day, with the follow schedule: A-1: 48 h; A-2: 36 h; A-3: 24 h; A-4: 12 h and on last day A-5: 1 h before exposition to behavioral tests (TST, FST). It is important to mention that the administration A-1, A-3 and A-5 were made 15 min before to exposition to CRS.

Later, having assessed the behavioral tests, the analysis of gastroprotective effect was based on the following procedure: Animals were sacrificed by overdose of sodium pentobarbital (PiSA, Agropecuaria, Guadalajara, Mexico) and the stomachs were removed, the mucosa was exposed by cutting along the greater curvature, and rinsed with normal saline solution and each stomach image was photographed with a digital camera (Canon EOS 70D (W), Tokyo, Japan). The area of mucosal injury was measured using ImageJ 1.44p software (National Institutes of Health, Bethesda, MD, USA) [40].

The ulcers were classified as level I, ulcer area <1 mm^2^; level II, ulcer area 1–3 mm^2^; and level III, ulcer area >3 mm^2^. The following parameters were determined: Ulcer index (UI) as 1x (number of ulcers level I) + 2x (number of ulcers level II) + 3x (number of ulcers level III); total area of lesion; percentage of lesion area in relation to total stomach area [41]) and the inhibition percentage was calculated using the following Formula [19]:$$Inhibition\;percentage=\frac{\begin{array}{c} \;TAL\;nontreated-TAL\;treated \end{array}}{TAL\;nontreated}\;\times 100$$

* TAL: *total area lesion*

#### 3.6.2. Tail Suspension Test (TST)

The TST has become one of the most widely used models for assessing antidepressant-like activity in mice. The test is based on the fact that animals subjected to the short-term, inescapable stress of being suspended by their tail, will develop an immobile posture. Mice were suspended on the edge of a Table 50 cm above the floor by adhesive tape placed approximately 1 cm from the tip of the tail. Immobility time was recorded over a 6 min period [42].

#### 3.6.3. Forced Swim Test (FST)

This is a widely used animal model for assessing antidepressant activity. Mice were placed in a Plexiglass cylinder (20 cm in height × 12 cm in diameter) filled with water (75 percent of total volume, at 24 ± 1 °C). In the pre-test session, each animal was subjected to the forced swim for 15 min, 24 h before the test, and during the test session, immobility time was recorded [43]. It was suggested that immobility reflects a state of lowered mood in which the animals have given up hope of finding an exit and have resigned themselves to the experimental situation [44].

### 3.7. Statistical Analysis

Data were analyzed with IBM^®^ SPSS Statistics^®^ Version 20.0 software (New York, NY, USA). All results were expressed as mean and standard error of the mean (SEM) and *p* < 0.05 was considered to be statistically significant. The statistical analysis between groups was carried out by one-way analysis of variance (ANOVA) followed by Bonferroni post hoc test.

## 4. Conclusions

Pharmacological analysis of *C. tenuiflora* allowed the identification of two antidepressant and gastroprotective fractions. FCt-2 and FCt-3, evaluated at 100 mg/kg. The major compounds of these bioactive fractions were: verbascoside, teniufloroside and a mixture of geniposide/musseanoside. All of these secondary metabolites displayed gastroprotective effects and antidepressant activity at 1 or 2 mg/kg. Although verbascoside was the only polyphenol that showed antidepressant activity in behavioral tests (TST, FST), tenuifloroside and the mixture of iridoids (GP/MS, 1 mg/kg) were also active in the FST test. In addition, this is the first time that misoprostol is reported as a gastroprotective drug that displayed a potent antidepressant effect; however, more studies are needed to clarify its mode of action.

## Figures and Tables

**Figure 1 molecules-24-01292-f001:**
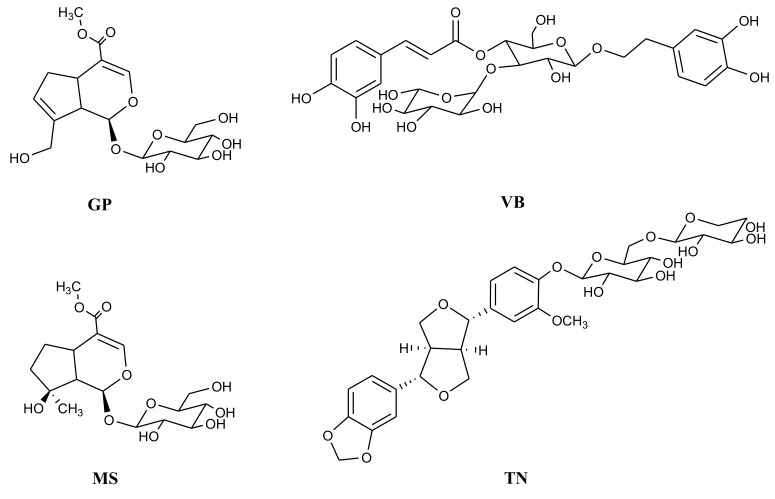
Bioactive compounds from *Castilleja tenuiflora* Benth. Genoposide (**GP**), mussaenoside (**MS**), verbascoside (**VB**) and tenuifloroside (**TN**).

**Figure 2 molecules-24-01292-f002:**
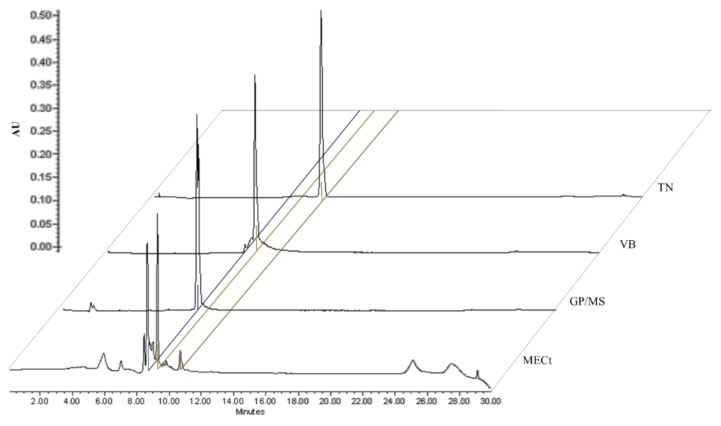
Chemical profiles of methanol extract (MECt), mixture of geniposide/mussaenoside (**GP**/**MS**, 8.3 min), verbascoside (**VB**, 9.15 min) and tenuifloroside (**TN**, 10.6 min). Chromatograms were developed at the UV wavelength of 240, 330 and 280 nm.

**Figure 3 molecules-24-01292-f003:**
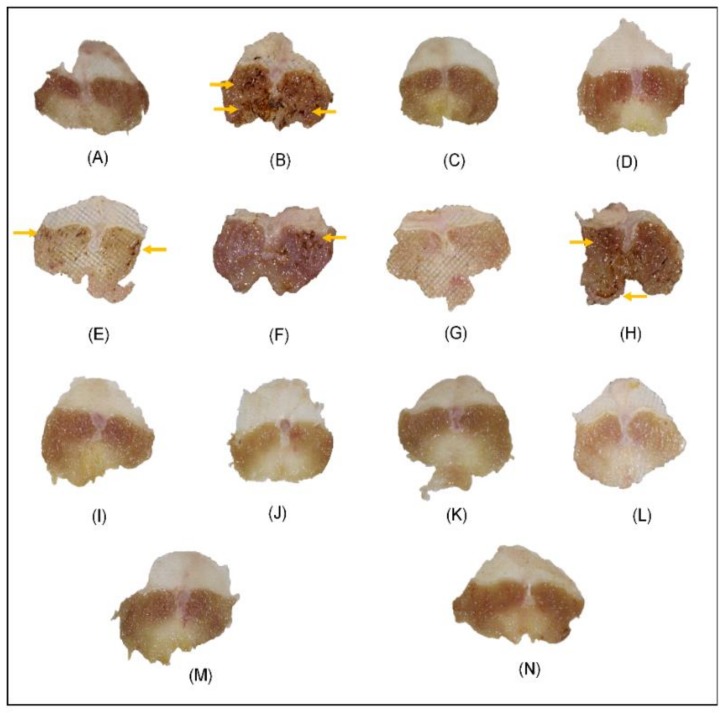
Macroscopic images of representative stomachs after CRS-induced gastric lesions in mice. Panels: (A) Basal; (B) VEH; (C) IMI; (D) MISO; (E) MECt; (F) FCt-2, (G) FCt-3 (H) FCt-4; and compounds at 1 mg/kg (I) **GP**/**MS**, (K) **VB**, (M) **TN**, and 2 mg/kg (J) **GP**/**MS**, (L) **VB**, (N) **TN**. Arrows indicate gastric ulcers.

**Figure 4 molecules-24-01292-f004:**
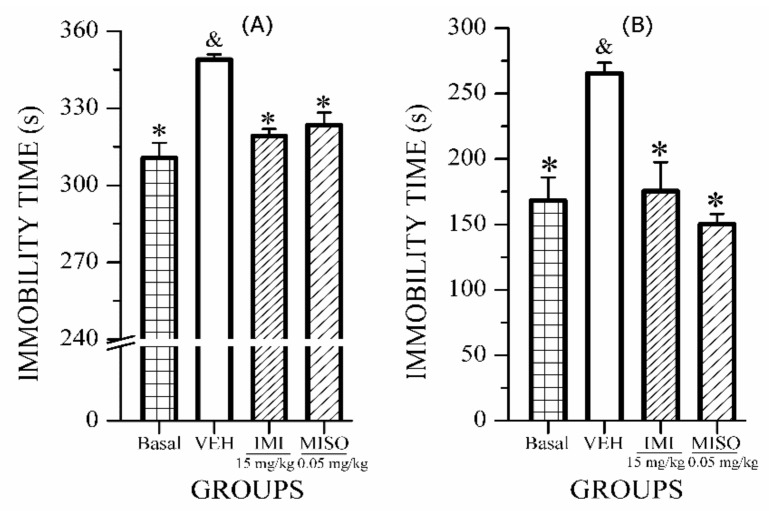
Effect of oral administration of drugs IMI and MISO on immobility time of ICR mice exposed to (**A**) TST and (**B**) FST. ANOVA followed by post hoc Bonferroni test (mean ± SEM) * *p* ≤ 0.05, in comparison with the VEH control group; & *p* ≤ 0.05 in comparison with the BASAL group.

**Figure 5 molecules-24-01292-f005:**
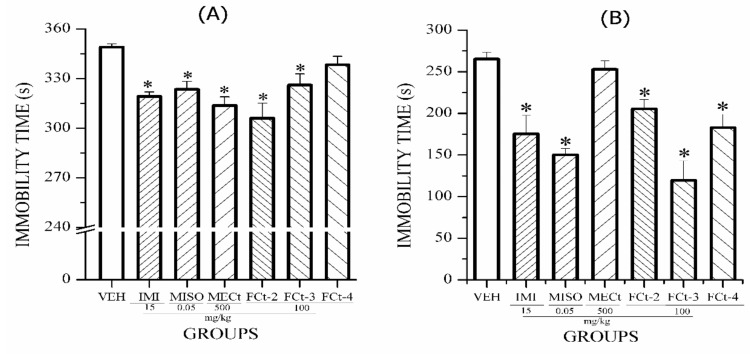
Effect of oral administration of MECt, FCt-2, FCt-3 and FCt-4 on immobility time of ICR mice during the tail suspension test (TST; (**A**)) and forced swim test (FST; (**B**)). ANOVA followed by post hoc Bonferroni test (mean ± SEM) * *p* ≤ 0.05, compared to the VEH control group.

**Figure 6 molecules-24-01292-f006:**
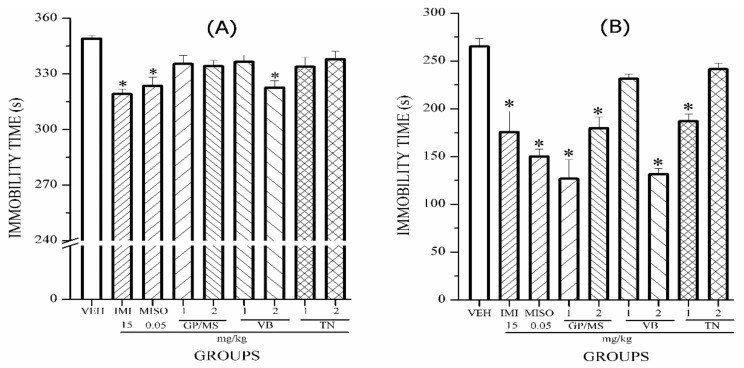
Effect of oral administration of the compounds geniposide/mussaenoside (**GP**/**MS**), verbascoside (**VB**) and tenuifloroside (**TN**) at 1 and 2 mg/kg on immobility time of ICR mice exposed to TST (**A**) and FST (**B**). ANOVA followed by post hoc Bonferroni test (mean ± SEM) * *p* ≤ 0.05, compared to the VEH control group.

**Figure 7 molecules-24-01292-f007:**
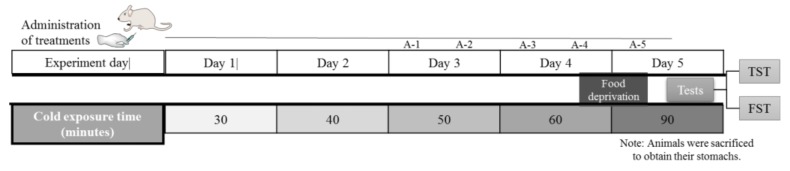
Chronology of the experimental design is modified from the sub-chronic stress model used in biological tests: Cold restraint stress (CRS), tail suspension test (TST) and forced swimming test (FST).

**Table 1 molecules-24-01292-t001:** Chemical content of **GP**/**MS**, **VB** and **TN** in each treatment from *Castilleja tenuiflora* Benth.

Doses (mg/kg)	Treatments	GP/MS	VB	TN
		mg/g of Treatment (mg kg^−1^ of Dose Administered)		
500	MECt	107.97 (53.99)	265.07 (132.54)	12.07 (6.04)
100	FCt-2	342.11 (34.21)	361.76 (36.18)	3.09 (0.31)
	FCt-3	42.43 (4.24)	153.93 (15.39)	11.42 (1.14)
	FCt-4	30.18 (3.02)	49.24 (4.92)	3.74 (0.37)

**Table 2 molecules-24-01292-t002:** Effect of treatments of *Castilleja tenuiflora* Benth. on mucosal lesions in the cold-restraint stress model.

Dose (mg/kg)	Group	Total Area Lesion (mm2)	% of Lesion Area	Ulcer Index	Inhibition (%)
	BASAL				
	VEH	47.36 ± 9.80	23.012 ± 8.34	16.5 ± 1.6	NA
15	IMI	0.36 ± 0.14 *	0.110 ± 0.05 *	1.5 ± 0.67	99.2
0.05	MISO	0.06 ± 0.01 *	0.017 ± 0.003 *	0.33 ± 0.2	99.8
500	MECt	1.10 ± 0.47 *	0.385 ± 0.172 *	4.1 ± 1.7	97.6
100	FCt-2	3.33 ± 1.38 *	1.854 ± 0.988 *	3.33 ± 1.2	92.9
	FCt-3	0.27 ±0.08 *	0.122 ± 0.047 *	2.8 ± 0.6	99.4
	FCt-4	18.11 ± 4.90 *	10.916 ± 4.36 *	12.6 ± 2.1	61.7
1	**GP/MS**	0.10 ± 0.04 *	0.026 ± 0.011 *	2.1 ± 0.4	99.7
	**VB**	0.05 ± 0.02 *	0.021 ± 0.002 *	0.66 ± 0.3	99.8
	**TN**	0.32 ± 0.12 *	0.089 ± 0.033 *	0.33 ± 0.2	99.3
2	**GP/MS**	0.10 ± 0.02 *	0.026 ± 0.008 *	0.83 ± 0.3	99.7
	**VB**	0.15 ± 0.03 *	0.045 ± 0.011 *	1.16 ± 0.4	99.6
	**TN**	0.48 ± 0.06 *	0.137 ± 0.015 *	1.0 ± 0.2	98.9

All values are expressed as mean ± S.E.M.; ANOVA followed by post hoc Bonferroni test * *p* ≤ 0.05, in comparison with the VEH control group.

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
