# Peer review of "In Vivo Gastroprotective and Antidepressant Effects of Iridoids, Verbascoside and Tenuifloroside from Castilleja tenuiflora Benth"

_molecules, 2019, doi:10.3390/molecules24071292_

Round 1

Reviewer 1 Report

Castilleja tenuiflora, a Mexican traditional medicine  for treatment of gastrointestinal diseases and nervous disorders was analysed to test their components. 1-check english language carefully.

2. I couldnt find  spectroscopic evidence of the pure compounds in the paper. please add this in the supplementary material.

3- mechanisms of gastroprotective activity is necessary nowadays, otherwise the paper is considered descriptive. 

4-hplc quantitation should de according  to some required proceedures  such as: The International Council for Harmonisation of Technical Requirements for Pharmaceuticals for Human Use (ICH)

Author Response

1. Castilleja tenuiflora, a Mexican traditional   medicine for treatment of gastrointestinal diseases and nervous   disorders was analyzed to test their components. 1-check english language   carefully

Response

 The English language has been checked and all changes were included in the final version of the manuscript

2. I couldn’t find spectroscopic evidence of the pure compounds in the paper. please add this in the supplementary material

Response

The spectroscopic data of isolated compounds were added in the text (section 3.1).

All spectra were included in the supplementary material.

3. mechanisms of gastroprotective activity is necessary nowadays, otherwise the paper is considered descriptive

Response

The research group have considered mechanistic studies for future analysis of the best treatments. 

4. hplc quantitation should be according to some required procedures such as: The International Council for Harmonization of Technical Requirements for Pharmaceuticals for Human Use (ICH)

Response

Authors are complete in agreement with this suggestion. In fact, we have the intention to develop the pharmacokinetic analysis of the best treatment where this analytical procedure will be included.  

In order to improve the quality of this manuscript, several analytical parameters (linearity, limit of detection and limit of quantification) were added in section 3.4.

Reviewer 2 Report

The article by Ricardo López-Rodríguez et al. is devoted to the development of new agents of a plant origin with antidepressant and potent anti-ulcerogenic effects. Now the task is of great importance in view of the widespread incidence of depression and its related psychiatric and somatic disorders. The plants considered to be a source of CNS-active substances with low toxicity and concomitant effects, which can be useful for prevention and treatment of depressive patients. The presented work continues their previous study of Castilleja tenuiflora Benth extracts and reveals a new data about its composition and pharmacological activity.  The novelty of work is that the authors have isolated and identified the most bioactive compounds of the extract fractions  - verbascoside, teniufloroside and mixture of geniposide/musseanoside. All substances proved to possess significant antidepressant activity and positive anti-ulcerogenic effect.

In pharmacological part of study the authors used the standard   models of stomach ulcer of animals induced by combination of cold and emotion stresses. The antidepressant effect of agents was evaluated by well known Porsolt test (forced swimming test) and less used tail suspension test. The effective antidepressant imipramine  and potent  gastroprotector misoprostol were used as the reference drugs. Thus, the presented work contains new data of scientific interest. The experimental design is adequate to the objectives of the study. The obtained results are confirmed by literature data of the relationship between the emotional stress and gastric ulceration and the early results about  pharmacological  activity of Castilleja tenuiflora Benth.

There are some questions and remarks to the work:

1.   The description of the method of ulcerative lesions evaluation does not contain a quantitative and qualitative analysis of damage to the gastric mucosa. Stomach injuries, as usual, include hyperemia (or bloodless), erosions and ulcers (large and small) on the surface of mucosa. Did the authors take into account the different contribution of these factors when evaluate the anti-ulcer effect?

2.   So the results of the inhibition percentage calculation  (Table 2) are questionable and do not correlate to the lesion area data in the same table. For example, the area of ulceration in IMI group  6 times more than in MISO group, but their inhibition percentage data by only  0.6 percent.

3.   On what basis did the authors choose doses of the extract, fractions, and individual components? Is it possible to compare the effectiveness of individual compounds (verbascoside, teniufloroside and mixture geniposide/musseanoside) with that in the MECt and its fractions, taking into account the differences in their quantity (administered doses)?

4.   In section “Material and Method” there is no data on the time period between agent administration and treatment (before or after?). There is also no data on the time period between last treatment and animal sacrification. It should be better to point out how many times the agent was administered.

5.   Why did the authors use the different route of administration for imipramine and misoprostol?

    It seems the oral route would be more appropriate for misoprostol as reference gastroprotective drug.

    Otherwise, taking into account that imipramine has stimulating adrenergic effect on locomotor activity,  in contrast to misoprostol, that has the sedative, locomotor decreasing effect, the authors conclusion about the antidepressant activity of  misoprostol need to be much better clarified and interpreted.

Author Response

1. The description of the method of ulcerative lesions evaluation does not contain a quantitative and qualitative analysis of damage to the gastric mucosa. Stomach injuries, as usual, include hyperemia (or bloodless), erosions and ulcers (large and small) on the surface of mucosa. Did the authors take into account the different contribution of these factors when evaluate the anti-ulcer effect?

Response

Considering this recommendation, authors have decided to improve both qualitative and quantitative analysis of damage to the gastric mucosa. These data were added in table 2 (section 2.2.1) 

2. So the results of the inhibition percentage calculation (Table 2) are questionable and do not correlate to the lesion area data in the same table. For example, the area of ulceration in IMI group 6 times more than in MISO group, but their inhibition percentage data by only 0.6 percent

Response

The above suggested analyses, allowed to reduce these doubts.

3A.  On what basis did the authors choose doses of the extract, fractions, and individual components? 

Response

Extract and fractions doses were chosen from previous reports where antidepressant activity and gastroprotective analyses were done by separately for this species (Herrera-Ruiz. M., et al 2015 and Sánchez, P.M., et al 2013).

Isolated compounds doses were chosen as an average of several naturally occurring compounds that display antidepressant or sedative activities

Resveratrol (1.25 mg/kg)

Huang, W., et al. Piperine potentiates the antidepressant-like effect of trans-resveratrol: Involvement of monoaminergic system. Metab. Brain Dis. 2013, 28, 585–595.

Verbascoside (1 and 3 mg/kg)

Juliao, L. S., et al Flavones and phenylpropanoids from a sedative extract of Lantana trifolia L. Phytochemistry 2010, 71, 294–300.

3B. Is it possible to compare the effectiveness of individual compounds (verbascoside, teniufloroside and mixture geniposide/musseanoside) with that in the MECt and its fractions, taking into account the differences in their quantity (administered doses)?

Response

Certainly, it is not possible to compare the effectiveness of the isolated compounds (1 and 2 mg/kg) vs complete extract (500 mg/kg) or its   fractions (100 mg/kg). However, these results (Table 2), allowed us the found that the isolated compound displayed both antidepressant activity and gastroprotective effect

4. In section “Material and Method” there is no data on the time period between agent administration and treatment (before or after?). There is also no data on the time period between last treatment and animal sacrification. It should be better to point out how many times the agent was administered

Response

Authors are very grateful with this observation. These missing data were included in the description of figure 7. Section 3.5.1.

5.  Why did the authors use the different route of administration for imipramine and misoprostol?

It seems the oral route would be more appropriate for misoprostol as reference gastroprotective drug.

Otherwise, taking into account that imipramine has stimulating adrenergic effect on locomotor activity, in contrast to misoprostol, that has the sedative, locomotor decreasing effect, the authors conclusion about the antidepressant activity of misoprostol need to be much better clarified and interpreted.

Response

MISO was administered by oral pathway and imipramine was administered intraperitonially.

In order to avoid this confusion, the paragraph was modified in the material and methods section.

“All C. tenuiflora treatments, Tween 20 (TW 1%, Merck, vehicle) were administrated by oral pathway (o.p) and only imipramine was intraperitonially (i.p.)”.

Round 2

Reviewer 1 Report

THE MANUSCRIPT HAS BEEN IMPROVED AND CAN BE ACCEPTED IN THE PRESENT FORM